# First-Line Chemoimmunotherapy versus Sequential Platinum-Based Chemotherapy Followed by Immunotherapy in Patients with Non-Small Cell Lung Cancer with ≤49% Programmed Death-Ligand 1 Expression: A Real-World Multicenter Retrospective Study

**DOI:** 10.3390/cancers15204988

**Published:** 2023-10-14

**Authors:** Keiko Tanimura, Takayuki Takeda, Nobutaka Kataoka, Akihiro Yoshimura, Kentaro Nakanishi, Yuta Yamanaka, Hiroshige Yoshioka, Ryoichi Honda, Kiyoaki Uryu, Mototaka Fukui, Yusuke Chihara, Shota Takei, Hayato Kawachi, Tadaaki Yamada, Nobuyo Tamiya, Naoko Okura, Takahiro Yamada, Junji Murai, Shinsuke Shiotsu, Takayasu Kurata, Koichi Takayama

**Affiliations:** 1Department of Respiratory Medicine, Japanese Red Cross Kyoto Daini Hospital, Kyoto 602-8026, Japan; keiko-t@koto.kpu-m.ac.jp (K.T.); nkataoka@koto.kpu-m.ac.jp (N.K.); aki-y@koto.kpu-m.ac.jp (A.Y.); 2Department of Thoracic Oncology, Kansai Medical University Hospital, Hirakata 573-1191, Japan; nakanisk@hirakata.kmu.ac.jp (K.N.); yuta1106yamanayt@gmail.com (Y.Y.); hgyoshioka@gmail.com (H.Y.); kuratat@hirakata.kmu.ac.jp (T.K.); 3Department of Respiratory Medicine, Asahi General Hospital, Asahi 289-2511, Japan; honda_r1@icloud.com; 4Department of Respiratory Medicine, Yao Tokushukai General Hospital, Yao 581-0011, Japan; kiyoaki.uryuu@tokushukai.jp; 5Department of Respiratory Medicine, Uji-Tokushukai Medical Center, Uji 611-0041, Japan; fukuimo@koto.kpu-m.ac.jp (M.F.); c1981311@koto.kpu-m.ac.jp (Y.C.); 6Department of Pulmonary Medicine, Graduate School of Medical Science, Kyoto Prefectural University of Medicine, Kyoto 602-8566, Japan; tki-sho@koto.kpu-m.ac.jp (S.T.); kwhat@koto.kpu-m.ac.jp (H.K.); tayamada@koto.kpu-m.ac.jp (T.Y.); takayama@koto.kpu-m.ac.jp (K.T.); 7Department of Respiratory Medicine, Rakuwakai Otowa Hospital, Kyoto 607-8062, Japan; koma@koto.kpu-m.ac.jp; 8Department of Respiratory Medicine, Matsushita Memorial Hospital, Moriguchi 570-8540, Japan; ku-n07@koto.kpu-m.ac.jp (N.O.); yamada.takahiro007@jp.panasonic.com (T.Y.); 9Department of Respiratory Medicine, Japanese Red Cross Kyoto Daiichi Hospital, Kyoto 605-0981, Japan; hanasaka@koto.kpu-m.ac.jp (J.M.); sshiotsu@gmail.com (S.S.)

**Keywords:** chemoimmunotherapy, first-line platinum-based chemotherapy, immune checkpoint inhibitor, non-small cell lung cancer, programmed death-ligand 1 expression

## Abstract

**Simple Summary:**

Chemoimmunotherapy (CIT) improved overall survival and progression-free survival (PFS) among patients with non-small cell lung cancer (NSCLC), while CIT showed limited efficacy among the subgroup with ≤49% programmed death-ligand 1 (PD-L1) expression. Therefore, sequential treatment with first-line platinum-based chemotherapy followed by immune checkpoint inhibitor treatment (SEQ) could be an option and was compared with CIT in this study. A total of 234 eligible patients with NSCLC with ≤49% PD-L1 expression from nine hospitals in Japan were analyzed. The median PFS in the CIT group (9.3 months (95% CI: 6.7–14.8)) was longer than SEQ group (5.5 months (95% CI: 4.5–6.1)) among the subgroup with 1–49% PD-L1 expression (*p* < 0.001). In contrast, no significant difference was observed among the <1% PD-L1 expression subgroup (*p* = 0.883). The median PFS-2 (the time from first-line treatment to progression to the second-line treatment or death) in the SEQ group was significantly longer than the median PFS in the CIT group (10.5 months (95% CI: 5.9–15.9) vs. 6.4 months (95% CI: 4.9–7.5); *p* = 0.024). Thus, CIT is recommended for patients with NSCLC with 1–49% PD-L1 expression.

**Abstract:**

Background: The long overall survival (OS) observed among patients with non-small cell lung cancer (NSCLC) with high programmed death-ligand 1 (PD-L1) expression in chemoimmunotherapy (CIT) groups in previous phase III trials suggests the limited efficacy of CIT among the subgroup with ≤49% PD-L1 expression on tumor cells. Hence, sequential treatment with first-line platinum-based chemotherapy followed by second-line immune checkpoint inhibitor treatment (SEQ) is an option. This study examined whether first-line CIT would provide better outcomes than SEQ in patients with advanced NSCLC with ≤49% PD-L1 expression. Methods: This retrospective study evaluated patients with untreated NSCLC who received first-line CIT or SEQ at nine hospitals in Japan. OS, progression-free survival (PFS), PFS-2 (the time from first-line treatment to progression to second-line treatment or death), and other related outcomes were evaluated between the CIT and SEQ groups. Results: Among the 305 enrolled patients, 234 eligible patients were analyzed: 165 in the CIT group and 69 in the SEQ group. The COX proportional hazards model suggested a significant interaction between PD-L1 expression and OS (*p* = 0.006). OS in the CIT group was significantly longer than that in the SEQ group in the 1–49% PD-L1 expression subgroup but not in the <1% PD-L1 expression subgroup. Among the subgroup with 1–49% PD-L1 expression, the CIT group exhibited longer median PFS than the SEQ group (CIT: 9.3 months (95% CI: 6.7–14.8) vs. SEQ:5.5 months (95% CI: 4.5–6.1); *p* < 0.001), while the median PFS in the CIT group was not statistically longer than the median PFS-2 in the SEQ group (*p* = 0.586). There was no significant difference between the median PFS in the CIT and SEQ groups among the <1% PD-L1 expression subgroup (*p* = 0.883); the median PFS-2 in the SEQ group was significantly longer than the median PFS in the CIT group (10.5 months (95% CI: 5.9–15.3) vs. 6.4 months (95% CI: 4.9–7.5); *p* = 0.024). Conclusions: CIT is recommended for patients with NSCLC with 1–49% PD-L1 expression because it significantly improved OS and PFS compared to SEQ. CIT had limited benefits in patients with <1% PD-L1 expression, and the median PFS-2 in the SEQ group was significantly longer than the median PFS in the CIT group. These findings will help physicians select the most suitable treatment option for patients with NSCLC, considering PD-L1 expressions.

## 1. Introduction

Lung cancer, with approximately 50% detected in advanced stages, is the leading cause of mortality among patients with cancer. However, the number of deaths has declined in recent years because of treatment progress [1]. Recent developments in the applications of immune checkpoint inhibitors (ICIs) have improved the prognosis of patients with advanced non-small cell lung cancer (NSCLC) without oncogenic driver genes. Nivolumab, a fully human anti-programmed cell death-1 (PD-1) antibody, was the first ICI that demonstrated improved overall survival (OS) compared to docetaxel in patients with squamous (CheckMate 017 trial) [2] and non-squamous (CheckMate 057 trial) [3] NSCLC that experienced disease progression during first-line platinum-based chemotherapy. Subsequently, an anti-PD-1 antibody, pembrolizumab (KEYNOTE-010 trial) [4], and an anti-programmed death-ligand 1 (PD-L1) antibody, atezolizumab (OAK trial) [5], were introduced as second-line treatments for NSCLC. PD-L1 expression on tumor cells (KEYNOTE-001 trial) [6], tumor mutation burden (TMB) [7], mismatch repair deficiency [8], and tumor-infiltrating lymphocytes (TIL) [9] are important biomarkers for predicting the response to anti-PD-1/PD-L1 antibodies.

The clinical importance of PD-L1 expression on tumor cells has been established and applied to decision-making regarding the first-line treatment of NSCLC. Pembrolizumab monotherapy in patients with previously untreated NSCLC with ≥50% PD-L1 expression demonstrated superiority in the median progression-free survival (PFS) and OS over the platinum-based chemotherapy (KEYNOTE-024 trial) [10]. The median OS in the pembrolizumab group was longer than that in the chemotherapy group, even after an in-study crossover to pembrolizumab in 55.0% of the platinum-based chemotherapy group [11]. Thus, pembrolizumab should be administered as first-line treatment in patients with ≥50% PD-L1 expression. In contrast, the median OS of pembrolizumab monotherapy in patients with 1–49% PD-L1 expression was similar to that of the chemotherapy group (KEYNOTE-042 trial) [12]. Similar results were obtained for first-line atezolizumab monotherapy compared with chemotherapy (IMpower110 trial) [13], in which higher PD-L1 expression was associated with a longer median OS. Therefore, the significance of ICI monotherapy in untreated NSCLC with 1–49% PD-L1 expression remains unclear.

Chemoimmunotherapy (CIT), combining platinum-based chemotherapy and ICIs, was developed to improve PFS and OS and prevent early disease progression during ICI monotherapy. CIT significantly improved PFS and OS compared to platinum-based chemotherapy, irrespective of the PD-L1 expression status, among patients with non-squamous NSCLC (KEYNOTE-189, IMpower150, and IMpower130 trials [14,15,16] and squamous NSCLC (KEYNOTE-407 trial) [17]. Although improved PFS and OS were observed in the CIT group compared to the platinum-based chemotherapy group across the stratified subgroups based on the PD-L1 expression status, a longer PFS was observed among patients treated with CIT with higher PD-L1 expression [18]. Furthermore, the median PFS-2, defined as the time from randomization to progression to next-line treatment or death, whichever occurred first, in the platinum-based chemotherapy group was equivalent to the median PFS in the CIT group in the KEYNOTE-189 and the KEYNOTE-407 trials [19,20].

These results indicate the efficacy of ICI therapy after progression to platinum-based chemotherapy. Thus, sequential treatment with first-line platinum-based chemotherapy followed by second-line ICI treatment (SEQ) is an important option in patients with ≤49% PD-L1 expression. This contrasts that of patients with ≥50% PD-L1 expression in which ICI administration (pembrolizumab monotherapy or as CIT) is indispensable as first-line treatment [11,14,15,16,17]. In addition, a pooled analysis of nivolumab in patients with previously treated NSCLC with <1% PD-L1 expression revealed a four-year OS of 11% [12]; a five-year update of pembrolizumab in patients with previously treated NSCLC with 1–49% PD-L1 expression showed a four-year OS of 9.2% [21], both supporting sequential treatment.

Although several reports have compared the prognostic value of CIT and pembrolizumab monotherapy [22,23,24,25], reports comparing CIT and SEQ are scarce, with only one report on non-squamous NSCLC with a small sample size showing the non-inferiority of sequential treatment over CIT [26]. Thus, whether SEQ in patients with NSCLC with ≤49% PD-L1 expression is comparable to that of CIT is unclear.

Therefore, this multicenter retrospective study examined whether first-line CIT for NSCLC would provide better OS and PFS than SEQ in advanced NSCLC with PD-L1 expression of ≤49%. In this study, the difference between PFS in CIT and PFS-2 in SEQ was also analyzed because CIT included platinum-based chemotherapy and ICI, which were administered throughout the PFS-2 period in the SEQ group. Subsequently, the association between subsequent treatment with CIT or SEQ and OS was investigated.

## 2. Patients and Methods

### 2.1. Data Collection

This multicenter retrospective study evaluated consecutive Japanese patients with untreated NSCLC who received first-line CIT or SEQ between January 2016 and September 2021 at nine hospitals in Japan. The study protocol was approved by the Ethics Committees of the Japanese Red Cross Kyoto Daini Hospital (2 February 2022; S2021-43) and each participating hospital. The requirement for informed consent was waived due to the study’s retrospective nature. However, patients were allowed to opt out of the use of their data, and relevant information concerning the study was made available on each hospital’s website.

### 2.2. Patients

The eligibility criteria were as follows: age ≥ 20 years; pathologically diagnosed NSCLC without driver gene alteration; ≤49% PD-L1 expression (determined using the 22C3 antibody; Dako North America, Carpinteria, CA/Agilent Technologies, Santa Clara, CA, USA) on tumor cells; patients with evaluable lesions using response evaluation criteria in solid tumors (RECIST) version 1.1 [27]; and patients treated with CIT or SEQ as first-line treatment during the eligible period. Patients who received ICI monotherapy as third-line or subsequent treatment and patients with an Eastern Cooperative Oncology Group performance status (ECOG-PS) of ≥2 at the start of first-line treatment were excluded.

### 2.3. Response Evaluation and Outcome Assessment

OS was defined as the time from the start of treatment to death from any cause. PFS was defined as the time from the start of the first-line treatment to disease progression or death, whichever occurred first. PFS-2 in the SEQ group was defined as the time from the start of first-line platinum-based chemotherapy to disease progression following second-line ICI treatment or death, whichever occurred first. The objective response rate (ORR) and disease control rate (DCR) were defined as “the percentage of patients in the study or treatment group who achieved complete response (CR) or partial response (PR) to the treatment” and “the percentage of patients in the study or treatment group who have achieved CR, PR, and stable disease”, respectively [27]. A patient was not evaluable when no imaging or measurements were performed [27]. Adverse events were assessed in accordance with the Common Terminology Criteria for Adverse Events (version 5.0).

### 2.4. Statistical Analysis

OS and PFS curves were plotted using the Kaplan–Meier method. The log-rank test was used to evaluate the OS and PFS. Categorical variables were compared using Fisher’s exact test. Cox proportional hazards models were used for univariate or multivariate analyses of PFS and OS. Propensity score matching was performed for age, sex, smoking status, histology, PD-L1 expression, and clinical stages in a 1:1 ratio (caliper width: 0.2) between the CIT and SEQ groups. For all analyses, a *p*-value < 0.05 indicated statistical significance.

### 2.5. Software Tools

Statistical analyses were performed using GraphPad Prism8 (GraphPad Software, San Diego, CA, USA) and EZR statistical software version 1.55 (Saitama Medical Center, Jichi Medical University, Saitama, Japan), in addition to a graphical user interface for R (R Foundation for Statistical Computing, Vienna, Austria). The EZR statistical software is a modified version of the R commander designed to add statistical functions.

## 3. Results

### 3.1. Patients’ Backgrounds

Among the 305 patients enrolled, 31 patients with ECOG-PS of ≥2, 38 patients who received ICI monotherapy as the third- or later-line treatment, and 2 patients who received combination therapy including ipilimumab were excluded (Appendix A). Among the 234 patients analyzed, 165 and 69 were in the CIT and SEQ groups, respectively. The median follow-up period was significantly longer in the SEQ group (32.9 months (95% CI: 18.6–44.8)) than in the CIT group (19.1 months (95% CI: 16.7–22.0)) (*p* < 0.001). As shown in Table 1, the SEQ group showed a higher male predominance (91.3% vs. 78.2%, *p =* 0.016), more patients with squamous cell carcinoma (43.5% vs. 26.1%, *p* = 0.012), and more patients with 1–49% PD-L1 expression (79.7% vs. 57.0%, *p* = 0.001) than the CIT group. Propensity score matching (1:1, caliper 0.2) was performed to adjust for background factors between the two groups, and 138 patients (69 in each group) were extracted as the post-matched cohort (Appendix A).

### 3.2. OS in the CIT and SEQ Groups

In the entire cohort (*n* = 234), the median OS of the first-line treatment was 22.3 months (95% CI: 17.4–28.0) in the CIT group and 16.2 months (95% CI: 12.6–22.2) in the SEQ group, without statistically significant differences (*p* = 0.326) (Figure 1A). However, the median OS in the propensity score-matched cohort (*n* = 138) was 35.2 months (95% CI: 22.8-not reached (NR)) in the CIT group and 16.2 months (95% CI: 13.3–23.9) in the SEQ group, showing a significantly longer OS in the CIT group (*p* = 0.004) (Figure 1B).

The survival curves crossed approximately one year after the start of first-line treatment in the overall cohort, in contrast with the propensity score-matched cohort, where CIT showed consistently superior survival rates. Therefore, a COX proportional hazards model was used to evaluate the interactions between each background factor in the CIT and SEQ groups. The COX proportional hazards model showed that the HR for OS in the 1–49% PD-L1 expression subgroup was 0.53 (95% CI: 0.34–0.83), while the HR for OS in the <1% PD-L1 expression subgroup was 1.70 (95% CI: 0.82–3.51), suggesting a significant interaction for PD-L1 expression (*p* = 0.006) (Figure 2).

In the subgroup with 1–49% PD-L1 expression, the OS in the CIT group was significantly longer than that in the SEQ group (35.2 months (95% CI: 22.8-NR) vs. 15.3 months (95% CI: 12.5–22.2); *p* = 0.005) (Figure 3A). Contrastingly, the CIT group did not show superiority in OS over the SEQ group among the <1% PD-L1 expression subgroup (CIT: 11.5 months (95% CI: 9.4–16.5) vs. SEQ: 20.8 months (95% CI: 9.8–42.3); *p* = 0.146) (Figure 3B).

### 3.3. PD-L1 Expression-Stratified PFS in the CIT vs. PFS or PFS-2 in the SEQ Groups

Because a significant interaction of PD-L1 expression was elucidated, the difference between the PFS in the CIT and SEQ groups was evaluated based on ≤49% PD-L1 expression. In this analysis, the difference between PFS in the CIT and PFS-2 in the SEQ was also compared because platinum-based chemotherapy and ICI were administered throughout the PFS period in the CIT group and the PFS-2 period in the SEQ group (Figure 4A).

In the 1–49% PD-L1 expression subgroup, the CIT group exhibited longer median PFS than the SEQ group (Figure 4B; CIT: 9.3 months (95% CI: 6.7–14.8) vs. SEQ: 5.5 months (95% CI: 4.5–6.1); *p* < 0.001), while the median PFS in the CIT group was not statistically longer than the median PFS-2 in the SEQ group (Figure 4D; 9.7 months (95% CI: 8.1–13.3); *p* = 0.586). In contrast, there was no significant difference between the median PFS in the CIT and SEQ groups in the <1% PD-L1 expression subgroup (Figure 4C; CIT: 6.4 months (95% CI: 4.9–7.4) vs. SEQ: 6.5 months (95% CI: 3.7–13.7); *p* = 0.883). Furthermore, the median PFS-2 in the SEQ group was significantly longer than the median PFS in the CIT group among the <1% PD-L1 expression subgroup (Figure 4E; 10.5 months (95% CI: 5.9–15.3) vs. 6.4 months (95% CI: 4.9–7.5); *p* = 0.024).

### 3.4. The Difference in Responses to CIT, Platinum-Based Chemotherapy, and ICI between Subgroups with 1–49% and <1% PD-L1 Expressions

The difference in ORR was compared between subgroups with PD-L1 expression of 1–49% and <1%; the CIT group showed an ORR of 58.7% in the 1–49% PD-L1 expression subgroup and an ORR of 40% in the <1% expression subgroup, with the 1–49% expression subgroup showing a significantly higher response (*p* = 0.02) (Table 2). In contrast, first-line platinum-based chemotherapy and second-line ICI treatment in the SEQ group showed no statistically significant differences in ORR between the subgroups with PD-L1 expression values of 1–49% and <1% (first line, *p* = 0.244; second line *p* = 1) (Table 2).

### 3.5. Subsequent Treatment after Immunotherapy

The association between subsequent therapy and the prognosis of patients treated with CIT or SEQ was also investigated. Of the 143 patients in the CIT group who discontinued first-line CIT, 84 (59%) underwent subsequent treatment, including 40 (28%) treated with anti-vascular endothelial growth factor (VEGF) therapy. Of the 69 patients in the SEQ group, 38 (55%) received subsequent treatment after second-line ICI treatment, including 7 (10%) who were treated with anti-VEGF therapy (Appendix A).

The OS was evaluated according to the following treatment options: chemotherapy with anti-VEGF therapy, chemotherapy without anti-VEGF therapy, and no subsequent treatment. In the CIT group, patients who received chemotherapy with anti-VEGF therapy showed a longer OS (32.3 months (95% CI: 21.9-NR)) than those without anti-VEGF therapy (17.4 months (95% CI: 8.9–23.8)) and no subsequent chemotherapy (9.6 months (8.2–22.3), *p* = 0.002) (Appendix A). In contrast, there was no significant difference in OS among the three subgroups of the SEQ group (*p* = 0.128; Appendix A).

### 3.6. Adverse Events (AEs)

Grade 3 or 4 AEs occurred in 71 (43%) of the 165 patients in the CIT group and 22 (31.9%) of the 69 patients in the SEQ group, with no statistically significant difference between both groups (*p* = 0.143) (Appendix A). Hematologic toxicity of grade 3 or 4 AEs in the first-line treatment was more common in the CIT group (CIT: 53 (32.1%) vs. SEQ: 16 (23.2%)); however, the difference was not statistically significant (*p* = 0.209). Interstitial lung disease (ILD) occurred in 6 (8.7%) patients in the SEQ group during first-line platinum-based chemotherapy or second-line ICI treatment and 10 (6.1%) patients in the CIT group during first-line treatment, without a significant difference (*p* = 0.57).

## 4. Discussion

CIT has shown shorter PFS and OS in patients with ≤49% PD-L1 expression compared to patients with high PD-L1 expression [14,15,16]. Furthermore, in the KEYNOTE-189 and KEYNOTE-407 trials, the median PFS-2 in the platinum-based chemotherapy group was comparable to the median PFS in the CIT group [19,20]. Thus, first-line platinum-based chemotherapy followed by ICI monotherapy could be an option to avoid immune-related AEs during first-line treatment.

The current study is the first to compare the impact of CIT and SEQ on survival outcomes in patients with NSCLC with ≤49% PD-L1 expression. PD-L1 expression in NSCLC tumor cells is a pivotal factor influencing the cancer-immune set point [28]. While high PD-L1 expression suggests that the tumor is in the immune-inflamed phenotype, where ICI would work most effectively, low or no PD-L1 expression indicates that the tumor is in the immune-desert phenotype, which requires chemotherapy, radiotherapy, or an anti-CTL4 inhibitor to activate the priming phase [28]. Thus, NSCLC with low or no PD-L1 expression may not elicit an adequate antitumor response, and treatment with ICI monotherapy is not recommended.

The current study demonstrates that CIT significantly improved OS and PFS compared to SEQ among the 1–49% PD-L1 expression subgroup, which agrees with previous phase III trials [14,15,16,17] and the cancer-immune set point hypothesis of converting the immune-desert phenotype into an immune-inflamed phenotype by activating the priming phase with chemotherapy [28]. In contrast, CIT failed to improve OS and PFS compared with SEQ in the <1% PD-L1 expression subgroup. The different outcome in OS and PFS between <1% and 1–49% PD-L1 expression subgroups is one of the most important findings of the current study. Furthermore, the median PFS in the CIT group failed to surpass the median PFS-2 in the SEQ group among the 1–49% PD-L1 expression subgroup, which was another important finding. Because CIT includes platinum-based chemotherapy and ICI monotherapy, the PFS in CIT should surpass the PFS-2 in the SEQ group if CIT has a synergistic effect. If CIT exhibits an additive effect of platinum-based chemotherapy and ICI, SEQ would be a treatment option to avoid the AEs of CIT in some patients.

The mechanism underlying the current study’s findings can be partly explained by the effect of platinum-based chemotherapy on the tumor microenvironment. Cytotoxic chemotherapy exerts its anticancer effect by directly inhibiting tumor cell growth and apoptosis induction and affecting the tumor microenvironment in favor of ICI treatment. The release of damage-associated molecular patterns (DAMPs) from chemotherapy-killed cancer cells can induce immunogenic cell death [29]. Carboplatin, paclitaxel, and pemetrexed, which are used in CIT for NSCLC, promote antigen presentation to dendritic cells by translocating the endoplasmic reticulum-derived calreticulin to the cancer cell surface [30]. Moreover, cytotoxic chemotherapy changes the immune-desert tumor microenvironment into an immune-inflamed microenvironment [28], which is suitable for tumor immune responses by promoting the infiltration of effector T cells into the tumor via the upregulation of chemokine genes and antigen-presenting genes [31,32]. Therefore, the improved tumor microenvironment induced by chemotherapy during CIT enhances the activation of ICI’s immune responses. In contrast, with SEQ, ICIs are introduced during disease progression, when most tumor cells become resistant to chemotherapy. Platinum-resistant tumors are in a cold tumor state where TILs are reduced [28,33], attenuating effector cell activity in subsequent ICI treatment. Thus, CIT enhances anticancer immune responses before the development of chemotherapy resistance in the immunosuppressive tumor microenvironment.

CIT showed no advantage over SEQ in the <1% PD-L1 expression subgroup. This result could be explained by the adaptive regulation of PD-L1 expression on tumor and immune cells by IFN-γ [34]. The PD-L1 expressions on dendritic and tumor cells are induced by exposure to IFN-γ, which is released by effector T-cells [35], and the PD-L1 expression on immune cells is associated with the extent of effector T-cell associated gene expression, including *IFNG*, *GZMB*, and *CXCL9* [34], suggesting that those tumors without PD-L1 expression are in the hypoactive effector T-cell state. Thus, tumors without PD-L1 expression suffer from predominant immune escape mechanisms in the state of an extremely immune-desert phenotype, which cannot be altered into an immune-inflamed phenotype by chemotherapy alone. The CIT group with <1% PD-L1 expression showed a statistically lower ORR than the 1–49% PD-L1 expression subgroup in this study (58.7% vs. 40.0%; *p* = 0.02). Therefore, the activation of the immune response to PD-1/PD-L1 inhibition is not expected under such conditions, resulting in a poor outcome of CIT. Under such conditions, an anti-cytotoxic T-lymphocyte-associated protein 4 (CTLA-4) antibody, which strongly activates the priming phase in combination with PD-1 antibodies, promotes the sensitization and activation of antigen-specific T cells [36]. Anti-CTLA-4 and anti-PD-1/PD-L1 antibodies demonstrated antitumor efficacy irrespective of PD-L1 expression in tumor cells in three phase III trials [37,38,39]. Therefore, the SEQ strategy is an option for NSCLC with <1% PD-L1 expression when the anti-CTLA-4 antibody is not suitable for immune-related adverse events. Because patients with <1% PD-L1 expression comprise 66.3% of East-Asian patients with NSCLC [40], this finding would be helpful in the decision making of this population.

Concerning the subsequent treatment, chemotherapy with anti-VEGF therapy in the CIT group showed a longer OS than chemotherapy without anti-VEGF therapy, consistent with previous reports [41,42].

The current study had several limitations. First, this retrospective observational study with a limited sample size may have been susceptible to selection bias and other variables. The higher predominance with squamous cell carcinoma in the SEQ group could be a source for potential bias. Despite the attempts to correct them by propensity score matching, the influence of unknown variables may not have been fully eliminated. Second, the SEQ group did not include patients who deteriorated rapidly after first-line treatment and those who did not undergo second-line treatment with ICI. Thus, patients with a relatively favorable prognosis may have been included in the SEQ group compared to those in the CIT group. In previous reports, approximately 8% of the patients failed to switch to second-line treatment after first-line treatment, suggesting that the SEQ strategy for these patients would jeopardize the opportunity for ICI treatment [26]. Third, the difference in the follow-up period and subsequent treatment options may have affected the outcomes since CIT was approved approximately two years later than second-line ICI monotherapy.

## 5. Conclusions

In conclusion, CIT is recommended for patients with NSCLC with 1–49% PD-L1 expression because it showed a significantly longer median PFS and OS than SEQ. On the other hand, CIT had limited benefits in patients with <1% PD-L1 expression, and the median PFS-2 in the SEQ group was significantly longer than the median PFS in the CIT group. Thus, SEQ would be a treatment option to avoid the AEs of CIT in some patients. These findings will help physicians to select the most suitable treatment option for patients with NSCLC, considering PD-L1 expressions.

## Figures and Tables

**Figure 1 cancers-15-04988-f001:**
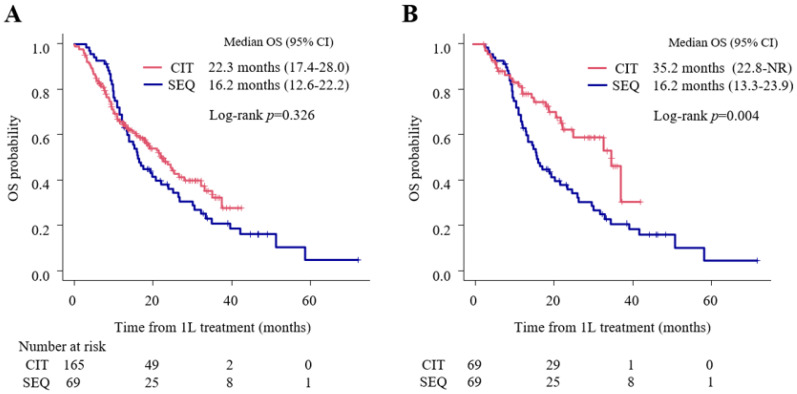
Kaplan–Meier estimates of overall survival (OS) in the overall cohort and the propensity score-matched cohort comparing the chemoimmunotherapy (CIT) and the sequential treatment with first-line platinum-based chemotherapy followed by second-line immune checkpoint inhibitor treatment (SEQ) groups. (**A**) Overall survival (OS) in the entire cohort (*n* = 234), (**B**) OS in the propensity score-matched cohort (*n* = 138, 1:1). CI, confidence interval; CIT, chemoimmunotherapy; 1L, first line; NR, not reached; OS, overall survival; SEQ, sequential treatment with first-line platinum-based chemotherapy followed by second-line immune checkpoint inhibitor treatment.

**Figure 2 cancers-15-04988-f002:**
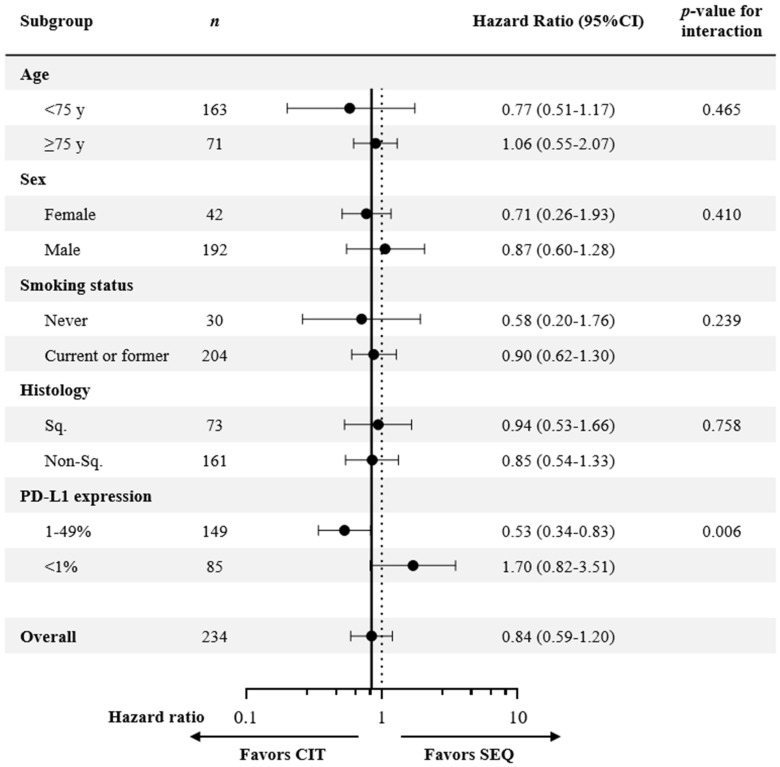
Subgroup analysis of Cox proportional hazards model of overall survival for chemoimmunotherapy (CIT) versus sequential treatment with first-line platinum-based chemotherapy followed by second-line immune checkpoint inhibitor treatment (SEQ) groups. CI, confidence interval; CIT, chemoimmunotherapy; PD-L1, programmed death-ligand 1; SEQ, sequential treatment with first-line platinum-based chemotherapy, followed by second-line immune checkpoint inhibitor treatment; Sq, squamous cell carcinoma.

**Figure 3 cancers-15-04988-f003:**
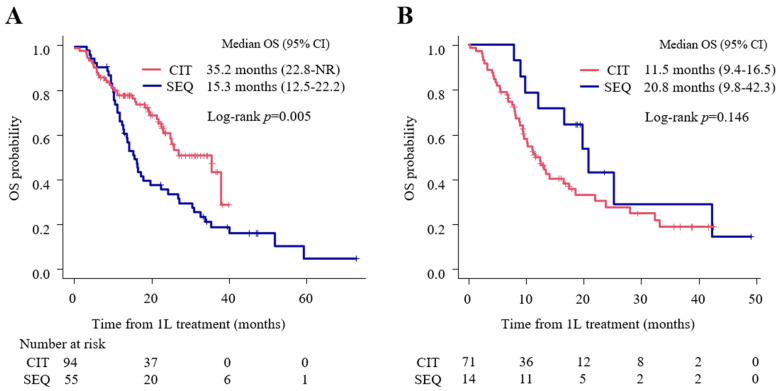
Kaplan–Meier analysis of overall survival (OS) in 1–49% and <1% PD-L1 expression subgroups compared between CIT and SEQ groups. (**A**) Overall survival (OS) in the 1–49% PD-L1 expression subgroup, (**B**) OS in the <1% PD-L1 expression subgroup. CI, confidence interval; CIT, chemoimmunotherapy; 1L, first-line; NR, not reached; OS, overall survival; SEQ, sequential treatment with first-line platinum-based chemotherapy followed by second-line immune checkpoint inhibitor treatment.

**Figure 4 cancers-15-04988-f004:**
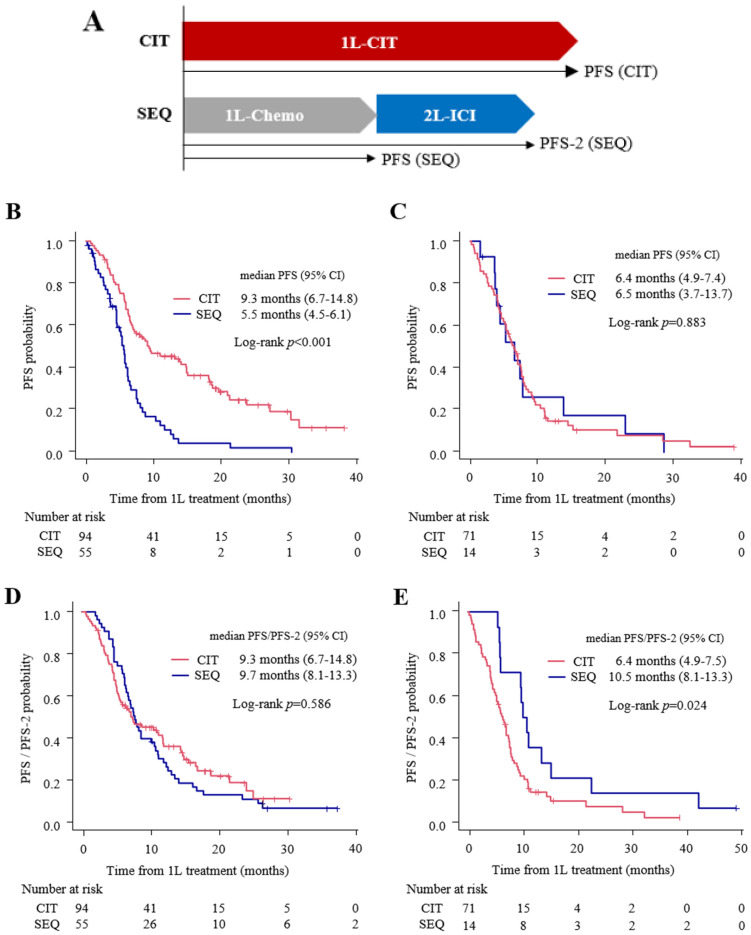
Kaplan–Meier estimates of progression-free survival (PFS)/PFS-2 in patients treated with CIT versus SEQ stratified by PD-L1 expression. (**A**) Definition of progression-free survival (PFS) and PFS-2 in this analysis. (**B**,**C**) PFS of the first-line treatment with CIT versus SEQ stratified by PD-L1 expression of 1–49% (**B**) or <1% (**C**). (**D**,**E**) PFS in patients treated with CIT versus PFS-2 in patients treated with SEQ stratified by PD-L1 expression of 1–49% (**D**) or <1% (**E**). *p* values were calculated using log-rank analysis. CI, confidence interval; CIT, chemoimmunotherapy; SEQ, sequential treatment with first-line platinum-based chemotherapy followed by second-line immune checkpoint inhibitor treatment; 1L, first-line; IO, immuno-oncology; PFS, progression-free survival; PD-L1, programmed death-ligand 1.

**Table 1 cancers-15-04988-t001:** Characteristics of patients enrolled in the CIT and SEQ groups in the overall population.

	*n* (%)	CIT (*n* = 165)	SEQ (*n* = 69)	*p*-Value
**Age (y),** **median (range):**		71 (44, 85)	70 (33, 82)	0.574
**Sex:**	Male	129 (78.2)	63 (91.3)	0.016
	Female	36 (21.8)	6 (8.7)	
**Smoking status:**	Never	24 (14.5)	6 (8.7)	0.285
	Current or former	141 (85.5)	63 (91.3)	
**Histology:**	Squamous cell	43 (26.1)	30 (43.5)	0.012
	Adeno	109 (66.1)	32 (46.4)	
	NOS	5 (3.0)	5 (7.2)	
	Others	8 (4.8)	2 (2.9)	
**PD-L1 expression:**	1–49%	94 (57)	55 (79.7)	0.001
	<1%	71 (43)	14 (20.3)	
**Stage:**	Recurrence	39 (23.6)	14 (20.3)	0.490
	III	8 (4.8)	6 (8.7)	
	IV	118 (71.5)	49 (71.0)	
**1L-CIT/SEQ regimen:**	Platinum/PTX or nab-PTX	-	30 (43.5)	
	Platinum/PEM/BEV	-	16 (23.2)	
	Platinum/PEM	-	9 (13.0)	
	Platinum/S-1	-	8 (11.6)	
	Others	-	6 (8.7)	
	Platinum/PEM/pembrolizumab	95 (57.6)	-	
	CBDCA/nab-PTX/pembrolizumab	48 (29.1)	-	
	CBDCA/nab-PTX/atezolizumab	11 (6.7)	-	
	CBDCA/PTX/BEV/atezolizumab	7 (4.2)	-	
	Platinum/PEM/atezolizumab	4 (2.4)	-	
**2L-ICI (SEQ):**	Pembrolizumab	-	32 (45.7)	
	Nivolumab	-	28 (40.0)	
	Atezolizumab	-	10 (14.3)	

BEV, bevacizumab; CBDCA, carboplatin; CIT, chemoimmunotherapy; 1L, first-line; ICI, immune checkpoint inhibitor; nab-PTX, nanoparticle albumin-bound paclitaxel; NOS, not otherwise specified; PD-L1, programmed death-ligand 1; PEM, pemetrexed; PTX, paclitaxel; S-1, Tegafur/Gimeracil/Oteracil; 2L, second line; SEQ, sequential treatment with first-line platinum-based chemotherapy followed by second-line immune checkpoint inhibitor treatment.

**Table 2 cancers-15-04988-t002:** Tumor response to 1L and 2L treatment with chemotherapy and immune checkpoint inhibitor according to PD-L1 expression status.

			PD-L1 Expression		
		*n* (%)	1–49%	<1%	*p*-Value
**CIT** (*n* = 165)	1L-CIT	ORR	58.7	40.0	0.02
		DCR	84.0	63.0	0.16
**SEQ** (*n* = 69)	1L-Chemo	ORR	45.5	26.7	0.244
		DCR	45.0	13.0	1
	2L-ICI	ORR	10.9	6.7	1
		DCR	40.0	46.7	0.769

Chemo, chemotherapy; CIT, chemoimmunotherapy; DCR, disease control rate; 1L, first line; ICI, immune checkpoint inhibitor; ORR, objective response rate; PD-L1, programmed death-ligand 1; 2L, second line; SEQ, sequential treatment with platinum-based chemotherapy followed by immune checkpoint inhibitors.

## Data Availability

The datasets analyzed in the current study are available from the corresponding author upon reasonable request.

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
