# Peer review of "First-Line Chemoimmunotherapy versus Sequential Platinum-Based Chemotherapy Followed by Immunotherapy in Patients with Non-Small Cell Lung Cancer with ≤49% Programmed Death-Ligand 1 Expression: A Real-World Multicenter Retrospective Study"

_cancers, 2023, doi:10.3390/cancers15204988_

Round 1
Reviewer 1 Report
excellent work.
line 128 is DAKO/AGILENT
THE ONLY THING TO ADD IS THE STATISTICS( IN JAPAN)OF PD-L1 POSITIVITY AND THE NUMBER OF PATIENTS/ECONOMICAL IMPACT IN CASE OF ACCEPTING YOUR CONCLUSIONS AS GUIDELINES
Author Response
I really appreciate your profound understanding of our manuscript. I also thank you for the important comments on our manuscript. I would like to provide the responses to your comments in a point-by-point manner.
line 128 is DAKO/AGILENT
Response:
I really appreciate your suggestion. I added ‘Agilent Technologies’ following “Dako North America’ in accordance with your suggestion.
THE ONLY THING TO ADD IS THE STATISTICS( IN JAPAN)OF PD-L1 POSITIVITY AND THE NUMBER OF PATIENTS/ECONOMICAL IMPACT IN CASE OF ACCEPTING YOUR CONCLUSIONS AS GUIDELINES
Response:
I really appreciate your advice referring to the distribution of PD-L1 expression in the real-world evidence. I added a reference (Respir Res. 2022) to show the proportion of patients with <1% PD-L1 expression. I added a sentence ‘Because patients with <1% PD-L1 expression comprise 66.3% of East-Asian patients with NSCLC (40), this finding would be helpful in the decision making of this population’ in the Discussion section (Lines 379-381).
Reviewer 2 Report
The authors conducted a retrospective study comparing the different sequences of the combination of ICIs and chemotherapy in NSCLC with PD-L1 <= 49%. It is an interesting study, and the conclusion is stable. However, there were several flaws in this manuscript.
Comment 1: The abstract is redundant. The content in lines 34 and 35 is repeated in lines 37 to 42.
Comment 2: The authors should provide detailed information about the process of propensity score matching, including matching by age, sex, stage, histology types, and other relevant factors.
Comment 3: In Figure 4, there were no instructions for C, D, and E. Please add the necessary information.
Author Response
I really appreciate deep understanding of our manuscript. I also thank you for the valuable comments on our manuscript. I would like to provide the responses to your comments in a point-by-point manner.
Comment 1: The abstract is redundant. The content in lines 34 and 35 is repeated in lines 37 to 42.
Response:
I really appreciate your suggestion. The PFS and PFS-2 were described in the Abstract, which made it difficult for the potential readers to understand the contents. Chemoimmunotherapy (CIT) includes platinum-based chemotherapy and ICI, which are administered throughout the PFS-2 period in the sequential administration (SEQ) group. Although I tried to add description to show them in the Abstract, I could not meet the word limit. Then, I did not change the Abstract.
Comment 2: The authors should provide detailed information about the process of propensity score matching, including matching by age, sex, stage, histology types, and other relevant factors.
Response:
I really appreciate your sincere advice. The information on the propensity score matching was described in the Result Section (Lines 168-171) in the original manuscript. I agree with you in that this statistical information should be described in the Patients and Methods. Then I added the description in the Patients and Methods section (2.4. Statistical analysis).
Comment 3: In Figure 4, there were no instructions for C, D, and E. Please add the necessary information.
Response:
I really appreciate your sincere advice. I am sorry for not providing the completed Figure Legends. I made a mistake in the description and corrected them in the revised manuscript (Lines 267-270).